# Development of Fluorescent Bacteria with *Lux* and Riboflavin Genes

**DOI:** 10.3390/ijms24065096

**Published:** 2023-03-07

**Authors:** Sun-Joo Lim, Miae Choi, Inseop Yun, Seulgi Lee, Ny Chang, Chan-Yong Lee

**Affiliations:** Department of Biochemistry, Chungnam National University, Daejeon 34134, Republic of Korea

**Keywords:** bioluminescence, fluorescence, *lux*, *Photobacterium*, riboflavin

## Abstract

Lumazine protein from marine luminescent bacteria of *Photobacterium* species bind with very high affinity to the fluorescent chromophore 6,7-dimethyl-8-ribitylumazine. The light emission of bacterial luminescent systems is used as a sensitive, rapid, and safe assay for an ever-increasing number of biological systems. Plasmid pRFN4, containing the genes encoding riboflavin from the *rib* operon of *Bacillus subtilis*, was designed for the overproduction of lumazine. To construct fluorescent bacteria for use as microbial sensors, novel recombinant plasmids (pRFN4-Pp N-*lum*P and pRFN4-Pp *lux*LP N-*lum*P) were constructed by amplifying the DNA encoding the N-*lum*P gene (*lux*L) from *P. phosphoreum* and the promoter region (*lux*LP) present upstream of the *lux* operon of the gene by PCR and ligating into the pRFN4-Pp N-*lum*P plasmid. A new recombinant plasmid, pRFN4-Pp *lux*LP-N-*lum*P, was constructed with the expectation that the fluorescence intensity would be further increased when transformed into *Escherichia coli*. When this plasmid was transformed into *E. coli* 43R, the fluorescence intensity of transformants was 500 times greater than that of *E. coli* alone. As a result, the recombinant plasmid in which the gene encoding N-LumP and DNA containing the *lux* promoter exhibited expression that was so high as to show fluorescence in single *E. coli* cells. The fluorescent bacterial systems developed in the present study using *lux* and riboflavin genes can be utilized in the future as biosensors with high sensitivity and rapid analysis times.

## 1. Introduction

Bioluminescence is the phenomenon of light emission resulting from enzyme-catalyzed oxidation reactions in living organisms. Light emission in marine bacteria catalyzed by bacterial luciferase involves the oxidation of long-chain fatty aldehydes and FMNH_2_, resulting in the emission of blue–green light [1,2,3] (Figure 1).
FMNH_2_ + RCHO + O_2_ → FMN + H_2_O + RCOOH + light

Genes encoding light-emitting enzymes and proteins exist on one operon, forming a group in bioluminescent bacteria. This *lux* operon consists of the *lux*AB gene encoding the α and β subunits of luciferase and the *lux*CDE genes encoding acyl-CoA reductase, acyl-transferase, and acyl-protein synthetase subunits of the fatty acid reductase complex (Figure 2) [4].

We found that various genes related to riboflavin (vitamin B_2_) biosynthesis exist downstream of the *lux* operon in *Photobacterium* spp. [5]. The biosynthesis of riboflavin is essential in bioluminescent bacteria because it is a precursor of FMNH_2_ (reduced flavin monophosphate), which is the substrate of the bacterial bioluminescence reaction, as described above. The lumazine protein identified in *P. phosphoreum* in 1970 [6] is a fluorescent protein that forms non-covalent bonds with 6,7-dimethyl-8-ribityllumazine (lumazine) [7,8]. The *lux*L gene, which encodes the lumazine protein, exists upstream of the *lux* operon in *Photobacterium* spp. (Figure 2), and the fact that the first gene (*rib*E) adjacent to *lux*G encodes riboflavin synthase draws molecular genetic interest. As the lumazine protein exhibits approximately 30% amino acid homology with riboflavin synthase [8,9] (Figure 3), it is assumed that the *lux*L gene was derived from a gene duplication of the riboflavin synthase gene (*rib*E) as a member of the riboflavin synthase superfamily [9].

**Figure 1 ijms-24-05096-f001:**
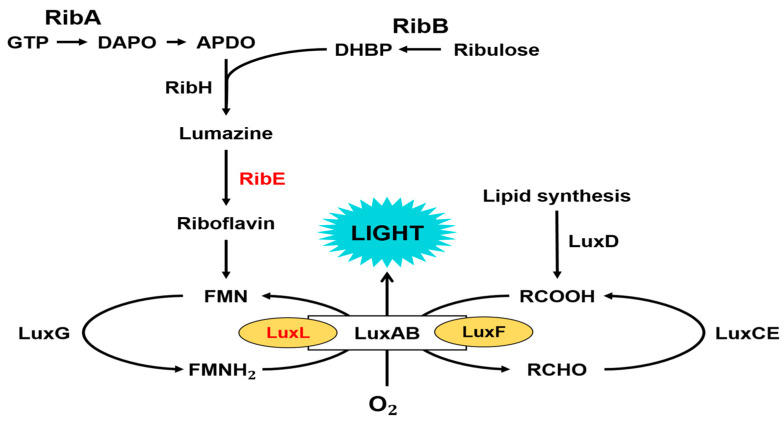
The complete catalytic reactions and proteins involved in light emission by *Photobacterium phosphoreum*. Proteins are as follows: lumazine protein (LuxL), α and β subunits of luciferase (LuxAB), fatty acid reductase complex (LuxCDE), non-fluorescent flavoprotein (LuxF), flavin reductase (LuxG), GTP cyclohydrolaseII (RibA), dihydroxy-butanone 4-phosphate synthase (RibB), lumazine synthase (RibH), and riboflavin synthase (RibE). Modified from reference [10].

**Figure 2 ijms-24-05096-f002:**
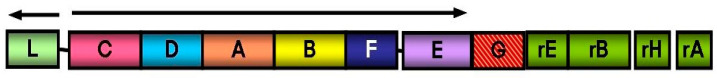
Gene organization of the *lux* operon region in bioluminescence bacteria of *Photobacterium phosphoreum* species. Arrows indicates the direction of transcription and “r” indicates riboflavin genes. The functions of genes are shown in Figure 1.

**Figure 3 ijms-24-05096-f003:**
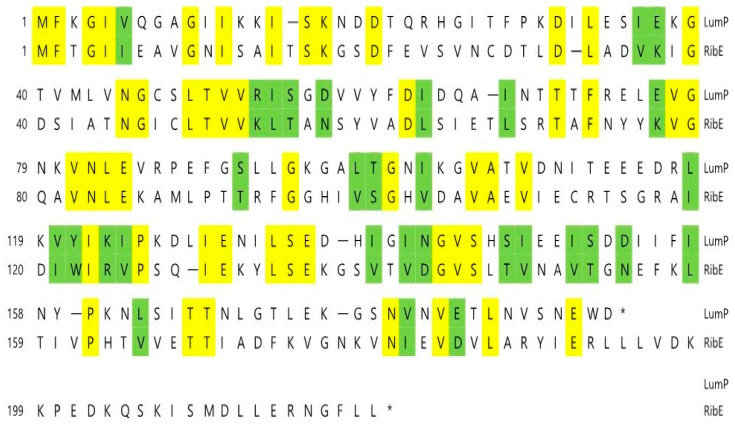
Comparison of amino acid sequences of lumazine protein (LumP) and riboflavin synthase (RibE) from *Photobacterium phosphoreum.* Identical or similar amino acids are shown boxed by yellow and green, respectively. Bars and asterisks indicate amino acid gaps and stop codons, respectively.

It is inferred that the lumazine protein functions as an optical transponder in light-emitting reactions of bioluminescent bacteria [9,11]. The lumazine protein shifts the wavelength and amplifies the maximum bioluminescence intensity in *Photobacterium* spp. by binding with lumazine, which is the fluorescent chromophore and substrate of riboflavin synthase [3,4]. The riboflavin synthase possesses the peculiar feature of intramolecular homology of the amino acid sequence between the N-terminal half domain (N-LumP) and C-terminal half domain (C-LumP) of the lumazine protein (Figure 4) [11], and it has been reported that lumazine binds only to N-LumP [12]. 

In this study, we employed plasmid pRFN4 [13], which contains five riboflavin biosynthesis genes of *B. subtilis* containing the mutated riboflavin synthase gene (*rib*B), for overproduction of lumazine. The nomenclature of the riboflavin biosynthetic genes in *B. subtilis* differs from those in other species. The *rib*B is the gene coding for riboflavin synthase from *B. subtilis* in the pRFN4 plasmid, whereas *rib*E is for riboflavin synthase in marine bioluminescent bacteria [5].

We inserted the gene encoding the N-terminal half domain of lumazine protein (N-LumP) into this plasmid to increase the fluorescence intensity, as lumazine binds to the lumazine protein with high affinity and functions as a chromophore (Figure 5). Moreover, we inserted intergenic DNA between *lux*L and the *lux*C promoter region (Figure 6) from *P. phosphoreum* to increase expression of the gene for lumazine protein (*lux*L). Intergenic DNA showed an extensive AT-rich region upstream of *P. phosphoreum lux*C. The GC content of this upstream region is less than half that of the *lux*-containing region, raising the possibility that the relative ease of melting DNA in AT-rich sequences may be related to the regulation of the *lux* system. We then transformed *E. coli* with a series of pRFN4 plasmids and conducted a spectroscopic study of the *E. coli* transformants, evaluating fluorescence intensity and examining the fluorescence of single cells using confocal microscopy.

## 2. Results

### 2.1. Construction of pRFN Plasmid Derivatives 

It has been reported [12] that Asn 101 and Ile 102 located beyond the N-terminal region are also involved in binding of the lumazine ligand with amino acids such as Ser 48, Thr 50, and Ala 66 at the binding sites in the N-terminal half of the lumazine protein (N-LumP) (Figure 3 and Figure 4). Therefore, the gene (*lux*L) encoding for N-LumP extending to the amino acid Glu 115 from *P. phosphoreum* was synthesized by PCR and ligated into the pRFN4 vector to construct the recombinant plasmid pRFN4-Pp N-*lum*P (Figure 5). We conducted the PCR process with the pT7-5 PpEE plasmid containing the *lux* genes of *P. phosphoreum* NCBM 844 using primers for amplifying the DNA of the gene encoding N-LumP, as well as DNA of the promoter regions of *lux*C and *lux*L by the PCR (Figure 6). We applied *EcoR*I and/or *Bam*H1 for restriction endonuclease digestion, purified the DNA product, and used T4 ligase for ligation into the pRFN4 plasmid [13], yielding plasmid pRFN4-Pp N-*lum*P, which contained the *lux*L gene coding for N-LumP.

We also performed PCR to insert *lux*C and *lux*L promoter regions (Figure 6), whose template was the pT7-5 PpEE plasmid containing the *P. phosphoreum lux* gene. To clone *lux*C and *lux*L promoters from *P. phosphoreum*, we performed the same PCR procedure as described above and constructed recombinant plasmids pRFN4-Pp *lux*CP and pRFN4-Pp *lux*CP-N-*lum*P. From these experiments, we generated recombinant plasmids pRFN4-Pp *lux*CP, pRFN4-Pp *lux*CP N-*lum*P, and pRFN4-Pp *lux*LP-N-*lum*P. The DNA inserts had sizes of approximately 9.0 kb, 9.4 kb, and 9.5 kb, respectively (Figure 5). Insertion of the correct DNA sequence was confirmed by DNA sequencing.

### 2.2. Fluorescence Intensities in E. coli

First, we examined the fluorescence intensity of *E. coli* cell extracts harboring the recombinant plasmid containing the gene encoding N-LumP from *P. phosphoreum.* The supernatant before sonication also exhibited fluorescence, indicating that the fluorophore was present in the liquid media. Fluorescence was monitored in the presence of lumazine with fixed excitation at 410 nm and riboflavin with excitation at 450 nm. To express the recombinant plasmids into which the *lux* gene and promoter DNA of *P. phosphoreum* were inserted, we transformed the plasmids into competent *E. coli* 43R cells and incubated them in LB (Luria–Bertani) medium. The incubated fluid was adjusted to a fixed volume and saturated with buffer solution A ((pH 7.2) consisting of 50 mM Tris-HCl (Tris-hydrochloride), 0.5 mM DTT (dithiothreitol), and 0.5 mM EDTA (ethylenediaminetetraacetic acid)). The supernatant of the mixture was obtained by ultrasonication and centrifugation, and the fluorescence intensity was evaluated using a fluorescence spectrophotometer.

#### 2.2.1. Fluorescence Intensities by Lumazine

A previous study [15] showed that among different *E. coli* strains, *E. coli* 43R [16] exhibited the most significant change in luminescence intensity of >1000-fold when transformed with a plasmid containing whole genes that are necessary for light emission from *P. leiognathi*. Based on the results of our recent study on the generation of fluorescent bacteria [5], we expected a larger increase in fluorescence intensity in *E. coli* 43R than in *E. coli* XL-1 when transformed with pRFN4 plasmids. First, we examined the emission spectrum of the transformed *E. coli* 43R at a fixed excitation wavelength of 410 nm (Figure 7a). In comparing the fluorescence intensities between *E. coli* 43R transformed with the pRFN4-Pp N-*lum*P plasmid and those transformed with the pRFN4 plasmid, the intensities increased by 5-fold. The fluorescence intensity in *E. coli* 43R transformed with the pRFN4-Pp *lux*LP-N-*lum*P plasmid, in which DNA of *lux*L promoter region was inserted into pRFN4-Pp N-*lum*P, was the highest among all *E. coli* transformed with pRFN4 plasmid derivatives (Figure 7a). 

The fluorescence intensity of *E. coli* 43R transformed with the pRFN4-Pp *lux*LP-N-*lum*P plasmid was 500-fold greater than that of *E. coli* 43R transformed with the pRFN4 plasmid, which may be due to non-covalent binding between the fluorescent lumazine protein and the lumazine chromophore. However, the insertion of the *lux*C promoter (pRFN4-Pp *lux*CP) or N-*lum*P (pRFN-Pp N-*lum*P) into pRFN4 did not increase the fluorescence intensity in *E. coli* cells transformed with these plasmids (Figure 7a). The fluorescence intensity of *E. coli* 43R transformed with recombinant plasmid pRFN4-Pp *lux*CP-N-*lum*P was lower than that of *E. coli* 43R transformed with pRFN4-Pp N-*lum*P plasmid (Figure 7a). Initially, we expected that the *lux*C promoter of *P. phosphoreum* would increase the fluorescence intensity by promoting the expression of the lumazine protein gene and by increasing binding of the lumazine ligand to N-LumP. However, the results showed that insertion of the *P. phosphoreum lux*C promoter did not increase the fluorescence intensity, suggesting that it failed to promote the expression of lumazine protein, as otherwise expected.

We also measured the excitation spectrum (Figure 7b) at a fixed emission wavelength of 490 nm by lumazine, in contrast to the previous process (Figure 7a). In accord with the results of the emission spectrum, the increase in the fluorescence intensity of the excitation spectrum was the highest in *E. coli* 43R transformed with the pRFN4-Pp *lux*LP-N-*lum*P plasmid among the transformed *E. coli* 43R with the pRFN4 plasmid derivatives. In particular, cell extract luminescence from 43R cells transformed with recombinant pRFN4-Pp *lux*LP-N-*lum*P was 100 times higher than that of *E. coli* 43R. Upon binding to lumazine, N-LumP had an absorbance maximum at 418 nm [12]; however, the excitation spectrum for emission at 490 nm of *E. coli* 43R cell extract transformed with pRFN4-Pp *lux*LP-N-*lum*P shows a maximum peak at approximately 360 nm. The different shapes of the excitation spectra compared to those of cells transfected with other plasmids, which have a maximum peak at approximately 418 nm, reflected that N-LumP binds to riboflavin as well as lumazine. 

Paulus et al. reported [17] that riboflavin, which possesses maximum absorption peaks at 450 and 370 nm, undergoes a substantial bathochromic shift (up to 20 nm) upon binding to N-LumP. The maximum absorption band at 410 nm originating from lumazine undergoes a bathochromic shift of approximately 5 nm upon binding of the protein. Moreover, the protein-bound ligand exhibited a broad, relatively weak absorbance band extending beyond 500 nm, whereas the free ligand was devoid of absorption [12].

#### 2.2.2. Fluorescence Intensities by Riboflavin

It is known that riboflavin competes with lumazine for binding to the lumazine protein [17]. Therefore, we examined the pattern and relative intensity of the emission spectra using the excitation wavelength of riboflavin at 450 nm from the cell extract transformed with pRFN4 plasmids (Figure 8a) and observed that the emission spectrum was similar to that of lumazine. In accord with the emission spectrum of the lumazine at an excitation wavelength of 410 nm, as shown in Figure 7a, it was observed that *E. coli* 43R transformed with the pRFN4-Pp *lux*LP-N-*lum*P plasmid has the strongest fluorescence intensity among the *E. coli* transformed with pRFN4 plasmid derivatives when the excitation wavelength was fixed at 450 nm (Figure 8a). *E. coli* 43R transformed with the pRFN4-Pp N-*lum*P plasmid showed approximately 2~3 times higher intensity than *E. coli* 43R transformed with the pRFN4 plasmid. Furthermore, the fluorescence intensity of *E. coli* 43R transformed with pRFN4-Pp *lux*LP-N-*lum*P was approximately 2~5 times greater than that of *E. coli* 43R transformed with the pRFN4 plasmid, yielding the strongest intensity among all transformed *E. coli* 43R (Figure 8 and Figure 9). 

We also measured the excitation spectrum for emission at a wavelength of 530 nm by riboflavin (Figure 8b), in contrast to the previous process of Figure 8a. This result is similar to that observed in results shown in Figure 7b, as the strength of the fluorescence intensities caused by riboflavin from the cell extracts transformed with the pRFN4-Pp *lux*LP-N-*lum*P plasmid were much stronger than that of *E. coli* 43R transformed with the pRFN4, pRFN4-Pp N-*lum*P, or pRFN4-Pp *lux*CP-N-*lum*P plasmids. However, in contrast to the fluorescence excitation spectrum of lumazine shown in Figure 7b, the excitation spectrum of riboflavin from *E. coli* 43R transformed with the pRFN4-Pp N-*lum*P plasmid showed the same pattern of fluorescence as *E. coli* 43R transformed with other pRFN4 plasmids, as shown in Figure 8b.

A comparison of fluorescence intensities from *E. coli* cell extracts transformed with different derivatives of pRFN plasmids is shown in Figure 9. The fluorescence intensity at 410 nm from the cell extract of *E. coli* 43R gradually increased by approximately 2∼6-fold after transformation with pRFN4 and insertion of the pRFN4-Pp N-*lum*P plasmid. Moreover, the increase in intensity of *E. coli* 43R was sharp and varied approximately 5∼6-fold when transformed with pRFN4-Pp *lux*LP-N-*lum*P (Figure 9). Therefore, the highest fluorescence intensity of cells containing recombinant plasmid pRFN4-Pp *lux*LP-N-*lum*P was 500 times higher compared to the intensity of *E. coli* 43R itself (Figure 9a).

However, the fluorescence spectroscopic results for *E. coli* 43R transformed with the recombinant plasmid which carried the *lux*C promoter of *P. phosphoreum* NCMB 844 was different from that of *E. coli* 43R transformed with the pRFN4 plasmid harboring the *lux* promoter of *P. leiognathi* described in our previous study [5]. The fluorescence intensity of *E. coli* 43R transformed with pRFN4-Pp *lux*CP or pRFN4-Pp *lux*CP-Pp N-*lum*P plasmids was higher than that of *E. coli* 43R transformed with the pRFN4 plasmid, but significantly lower than that of *E. coli* 43R transformed with the pRFN4-Pp N-*lum*P plasmid (Figure 9a,b).

The identities of lumazine and riboflavin in cell free extracts were finally confirmed by silica gel of TLC developed with 3% ammonium chloride. Lumazine and riboflavin were visualized by fluorescence upon excitation using a long-wavelength UV lamp. As shown in Figure 9c, lumazine emitted greenish–blue and riboflavin emitted yellowish–green light, respectively.

### 2.3. Single-Cell Fluorescence Imaging by Confocal Microscopy

We tested whether a single fluorescent *E. coli* cell could be detected using confocal microscopy. *E. coli* 43R cells transformed with the recombinant plasmid containing *lux* DNA from *Photobacterium* spp. was loaded into a small amount of LB medium. We examined single-cell images of fluorescent *E. coli* with DAPI (excitation 405 nm) and FITC (excitation 458 nm) filter sets in the super-resolution confocal laser scanning microscope LSM880 and additionally performed differential interference contrast (DIC) imaging and merged images. *E. coli* 43R itself without any plasmids did not show any fluorescence, whereas we observed fluorescence in single *E. coli* 43R cells transformed with derivatives of pRFN4 that contained DNA for the gene and its promoter. We inferred that the overproduction of lumazine ligand caused the transformed *E. coli* 43R to express fluorescence. The fluorescence of *E. coli* 43R transformed with recombinant pRFN4-Pp *lux*LP-N-*lum*P plasmid was greater than that of *E. coli* 43R transformed with the pRFN4 plasmid. Furthermore, in *E. coli* 43R cells transformed with the recombinant plasmid pRFN4-Pp *lux*LP-N-*lum*P that contained DNA of the *P. phosphoreum lux*L promoter region, the fluorescence was more intense, and more cells expressed green light, as shown in Figure 10.

## 3. Discussion

There has been a significant interest in luminescence and fluorescence for cellular and molecular imaging, and fluorescence is now the dominant methodology applied in biotechnology, flow cytometry, DNA sequencing, and genetic analysis [18,19]. The light emission of the bacterial luminescent system is being applied as sensitive, rapid, and safe assays in an ever-increasing number of biological systems and bioluminescence-based imaging of living cells has become an important tool in biological and medical research [20].

Our previous study [15] showed that transformed mutant *E. coli* RR1 emitted a drastically high level of light, which was strong enough to identify high luciferase enzyme activities in cell extracts, as well as to readily measure the fatty acid reductase activity responsible for supplying fatty aldehyde substrates. The light intensity was approximately 1000 times higher than that of the other transformed *E. coli* strains. This result is consistent with that observed for the *V. harveyi lux* system [16], indicating that a repressor or inhibitor may have been eliminated in the mutant *E. coli*. 

In a series of studies regarding the generation of fluorescent bacteria, we initially started to work with pRFN4 plasmids by inserting the lumazine protein gene from *P. leiognathi* to test fluorescence intensities and single-cell imaging [10]. By binding one molecule of lumazine to the protein, the fluorescence intensity of N-LumP is increased because of its lumazine chromophore [12]. In this study, we selected *P. phosphoreum,* which is the brightest strain among bioluminescent bacteria, to construct pRFN4 derivatives containing the *N-lum*P gene and/or its promoter region from *P. phosphoreum*. Accordingly, we expected that increased binding of the lumazine ligand with the lumazine protein would yield higher fluorescence, as the *lux*L promoter from *P. phosphoreum* promotes the expression of the N-terminal half of *lux*L gene encoding N-LumP. The fluorescence intensity was significantly increased by insertion of the *lux* promoter when *E. coli* was transformed with the recombinant plasmid pRFN4-Pp *lux*LP-N-*lum*P. Remarkably, when this plasmid was transformed into *E. coli* 43R, the fluorescence intensity of the transformant was 500 times higher than that of *E. coli* itself. 

Additionally, we conducted an imaging study with *E. coli* 43R transformed with a plasmid that harbored the lumazine protein gene of *P. phosphoreum* and a plasmid that was also inserted containing the *lux*L promoter region of *P. phosphoreum*. As shown in confocal microscopy images in Figure 10, the fluorescence of a single cell showed a similar tendency to the results from the fluorescence spectrophotometer. The single-cell image is in line with data from the fluorescence spectrophotometer, showing that the fluorescence of a single cell became stronger when *E. coli* was transformed with the recombinant plasmid containing the gene coding for N-LumP. As a result, cells transformed with the recombinant plasmid containing the gene coding for N-LumP as well as the *lux* promoter exhibited expression so high as to show fluorescence of single *E. coli* cells. Thus, the bacterial system with *lux* and riboflavin genes described in the present study can be utilized in the near future as a biosensor with high sensitivity and short analysis time.

Recently, bacterial bioluminescence has been used as a reporter system in plant protoplasts and single-cell imaging systems [20,21]. Microbial studies have many possibilities, such as mass production through incubation and improved selectivity by modulating incubation conditions to restrict unwanted metabolic pathways and induce adaptation to certain environmental conditions [20,21,22,23]. In addition, the *lux* biosensor based on *B. subtillis* for DNA-tropic and oxidative stress-causing agents was constructed [24]. Therefore, if future research continues to emphasize the possibilities for microbial studies and applications, the results presented in this paper will provide a solid basis for the development of microbial biosensors. 

## 4. Materials and Methods

### 4.1. Strains and Vectors 

We used *E. coli* XL-1 Blue as the cloning strain and measured fluorescence in *E. coli* 43R, an isogenic strain of *E. coli* RR1 [15]. The pRFN4 plasmid containing *B. subtilis* riboflavin biosynthetic genes was used as a cloning vector [13] (Table 1). 

### 4.2. Preparation of Recombinant Plasmids by PCR

DNA was amplified via PCR using the pT7-5 PpEE plasmid as a template. This plasmid contained the DNA of *lux*L and *lux*C and their promoter DNAs from *P. phosphoreum* NCBM 844 (Table 1). To amplify DNA containing the gene coding for the N-terminal half domain (N-LumP) up to the 115th amino acid (glutamic acid), as well as the *lux*L and *lux*C promoter regions shown in Figure 6, we used forward and reverse primers (Table 2). The PCR process consisted of 5 stages to duplicate pRFN4-Pp *lux*CP and pRFN4-Pp *lux*CP-N-*lum*P plasmids. The conditions of each stage were: pre-denaturation 95 °C, 5 min; denaturation 95 °C, 20 s; annealing 50 °C, 20 s; extension 72 °C, 30 s; and final extension 72 °C, 3 min. We performed the same PCR amplification and restriction endonuclease reactions using the same enzymes. After transformation, the colonies on LB agar plate were picked and incubated in a small amount of LB medium, with a small amount of DNA extracted using a miniprep kit.

We then loaded and electrophoresed the products on 1.0% agarose gels. After electrophoresis, we identified the recombinant plasmids with the pRFN4-Pp *lux*CP and the pRFN4-Pp *lux*CP-N-*lum*P plasmids. The sizes of the inserts were approximately 9.0 kb and 9.4 kb, respectively.Among the 5 PCR stages, the processes of denaturation, annealing, and extension were repeated for 30 cycles. After PCR, the amplified DNA was purified via gel extraction. The purified DNA products and vectors cleaved by the same restriction endonuclease enzymes (*EcoR*I/*Bam*H1) were then ligated into pRFN4 plasmids by T4 DNA ligase.

The ligated DNA was transferred into competent *E. coli* XL-1 cells. We incubated these colonies and extracted a small amount of DNA. We checked for transformation by performing PCR and applying restriction enzymes. Finally, we confirmed the orientation and insertion of the correct sequences using DNA sequencing. *E. coli* 43R cells were transformed with the pRFN4 derivative plasmids pRFN4-Pp N-*lum*P, pRFN4-Pp *lux*CP, pRFN4-Pp *lux*CP-N-*lum*P, and pRFN4-Pp *lux*LP-N-*lum*P (Table 1 and Figure 5), into which the N-*lum*P gene and/or *lux* promoters had been inserted in the previous process. The cells were then incubated under the following conditions: *E. coli XL-1 Blue* and *E. coli* 43R transformed with appropriate plasmids were incubated in LB medium containing 100 μg/mL of ampicillin at 37 °C. Incubated recombinant *E. coli* cells were adjusted to a fixed volume (A_660_ × Volume = 30 mL). Each strain was centrifuged at 2,500 rpm for 20 min to sediment the cells.

### 4.3. Buffer Solutions

We dissolved each cell extract in 25 mL of buffer solution A (pH (7.2) consisting of 50 mM Tris-HCl (Tris-hydrochloride), 0.5 mM DTT (dithiothreitol), and 0.5 mM EDTA (ethylenediaminetetraacetic acid)). The supernatant from each solution was collected via ultrasonication (3 × 30 s) and centrifugation.

### 4.4. Fluorescence Analysis

Using a LS 45 fluorescence spectrophotometer (PerkinElmer, Llantrisant, UK), we scanned the emission spectrum of each supernatant at fixed excitation wavelengths of 410 and 450 nm, and scanned the excitation spectra at emission wavelengths of 490 and 530 nm, respectively. Lumazine and riboflavin in cell free extracts were confirmed by silica gel (Baker, NJ, USA) of TLC developed with 3% ammonium chloride. The authentic lumazine was a gift from Prof. Fischer’s lab at the University of Hamburg and riboflavin was purchased from Sigma-Aldrich. The reference compounds of lumazine and riboflavin, as well as those from *E. coli*, were visualized by fluorescence upon excitation using a long-wavelength UV lamp. 

### 4.5. Imaging of Fluorescent E. coli under Confocal Microscopy

Poly-L-lysine was smeared onto glass slides and incubated for at least 1 h. We then placed the *E. coli* on the prepared glass slides and fixed them by placing a cover-slip on top. Single-cell images of fluorescent *E. coli* were examined using a super-resolution confocal laser scanning microscope LSM 880 with Airyscan (Carl Zeiss, Jena, Germany).

## 5. Conclusions

Using modified pRFN4 plasmids containing the *lux* gene promoter DNA from the brightest strain of bioluminescent bacteria, *P. phosphoreum,* and with the gene coding for the minimum version of the amino-terminal half of lumazine protein (N-LumP), as well as the mutant strain of *E. coli* 43R, we developed fluorescent bacteria with 500 times the fluorescence intensity of normal *E. coli*. We can acquire single-cell images of fluorescent bacteria, and this can be used for a microbial sensor which can detect heavy metals, environmental pollutants, and antibiotics by checking their effects on bacterial fluorescence intensities. 

## Figures and Tables

**Figure 4 ijms-24-05096-f004:**
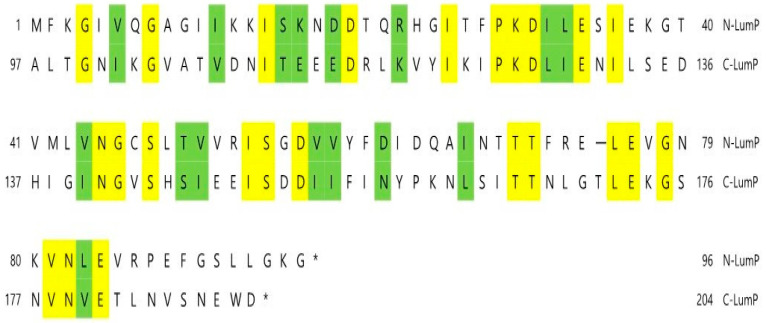
Amino acid sequence homology between the N-terminal half of lumazine protein (N-LumP) and the C-terminal half of lumazine protein (C-LumP) from *P. phosphoreum* NCMB844. Identical or similar amino acids are shown boxed by yellow and green, respectively. The bar and asterisks indicate amino acid gaps and stop codons, respectively.

**Figure 5 ijms-24-05096-f005:**
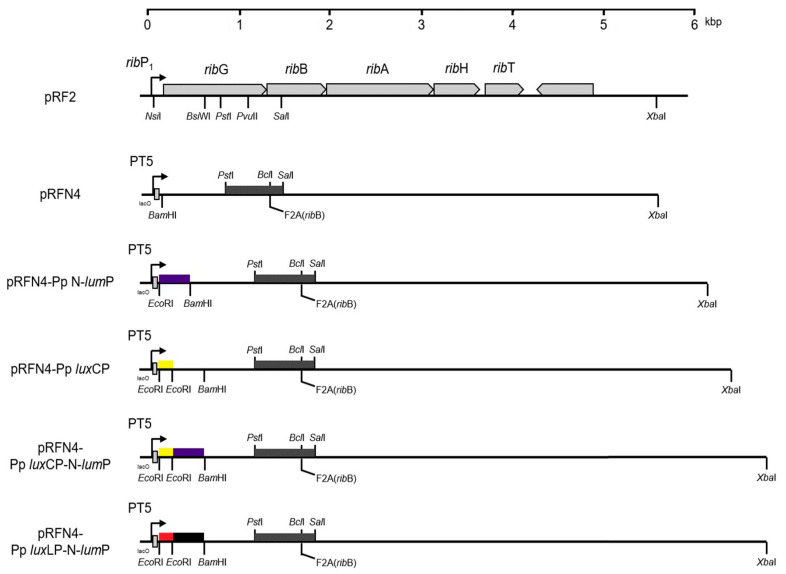
Gene map of recombinant plasmids with inserted *P. phosphoreum lux* genes. Recombinant pRFN4 plasmids used in this study: pRFN4, pRFN4-Pp N-*lum*P, pRFN4-Pp *lux*CP, pRFN4-Pp *lux*CP-N-*lum*P, and pRFN4-Pp *lux*LP-N-*lum*P plasmids. The nomenclature of riboflavin biosynthetic genes of *B. subtilis* differs from other species. Therefore, *rib*E in pRF2 and pRFN4 is the gene coding for riboflavin synthase in *B. subtilis* [5]. In pRFN4, the *rib*E gene coding for Phe 2 in riboflavin synthase was replaced with Ala for overproduction of lumazine (F2A in *rib*B) [13].

**Figure 6 ijms-24-05096-f006:**
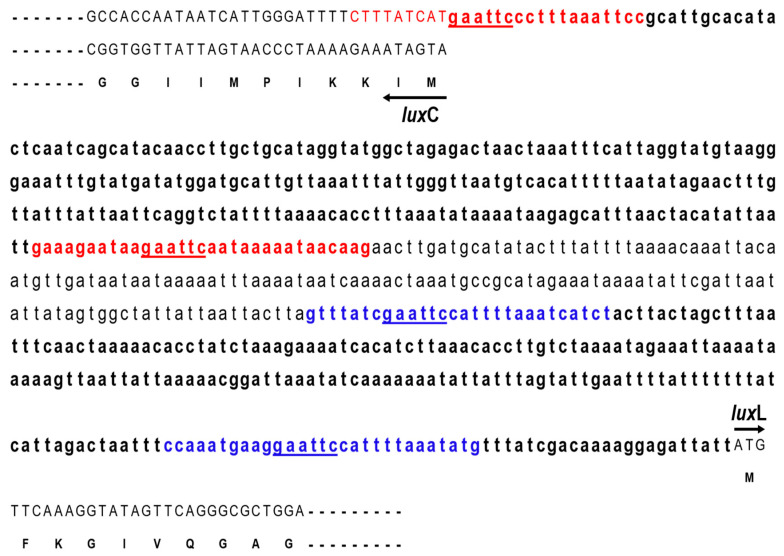
Nucleotide sequences of the intergenic region between *lux*L (*lum*P) and *lux*C genes of *P. phosphoreum*. PCR primer binding sequences of the *lux*L and *lux*C promoters are shown in blue and red colors, respectively. Restriction sites of *Eco*RI (GAATTC) are underlined. The *lux*C gene is transcribed by the complementary nucleotide sequences. The nucleotide sequence (accession number L21989) was reported in the author’s previous paper [14].

**Figure 7 ijms-24-05096-f007:**
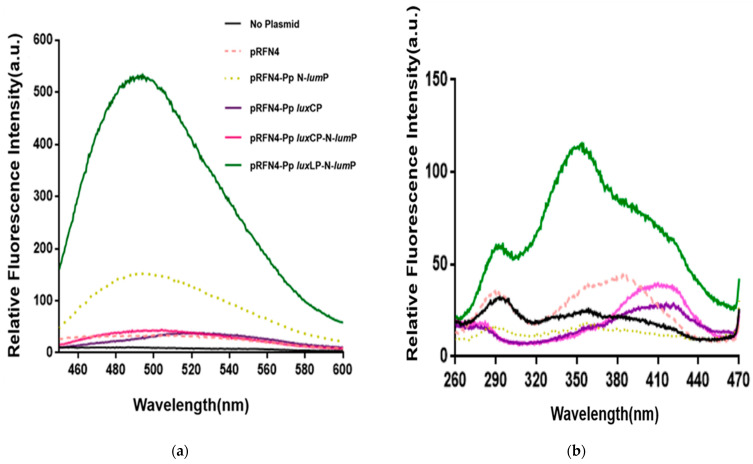
Fluorescence spectra of *E. coli* 43R transformed with recombinant pRFN4 plasmids containing *P. phosphoreum lux* DNA. (**a**) Emission spectrum in fixation at excitation wavelength of 410 nm. (**b**) Excitation spectrum for emission at 490 nm.

**Figure 8 ijms-24-05096-f008:**
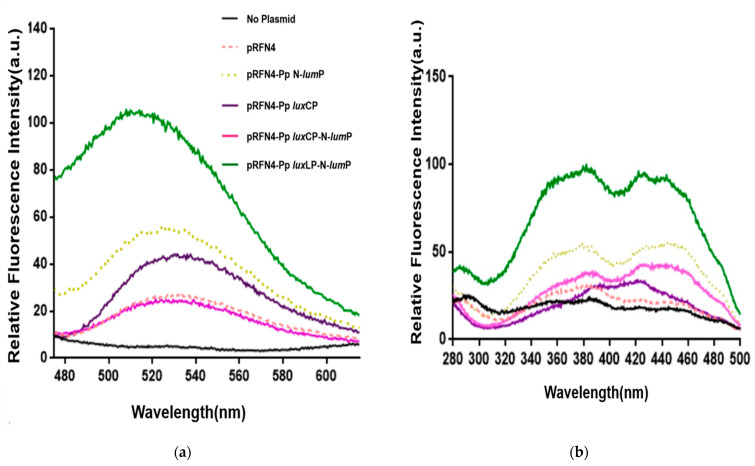
Fluorescence spectra of *E. coli* 43R transformed with recombinant pRFN4 plasmids containing *P. phosphoreum lux* DNA. (**a**) Emission spectra in fixation at excitation wavelength of 450 nm. (**b**) Excitation spectra for emission at 530 nm.

**Figure 9 ijms-24-05096-f009:**
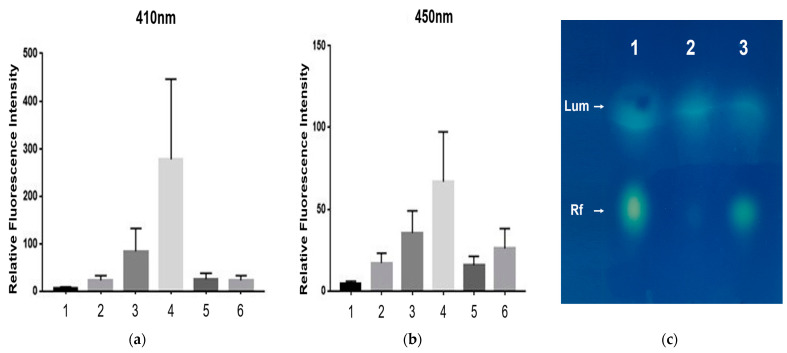
Relative fluorescence intensities. 1, *E. coli* 43R itself; 2, pRFN4 in *E. coli* 43R; 3, pRFN4-Pp N-*lum*P in *E. coli* 43R, 4, pRFN4-Pp *lux*LP-N-*lum*P in *E. coli* 43R; 5, pRFN4-Pp *lux*CP-N-*lum*P in *E. coli* 43R; 6, pRFN4-Pp *lux*CP in *E. coli* 43R. (**a**) Excitation wavelength was fixed at 410 nm for lumazine. (**b**) Excitation wavelength was fixed at 450 nm for riboflavin. (**c**) Identification of lumazine and riboflavin in TLC (thin layer chromatography) panel under UV illumination. 1, Mixtures of lumazine and riboflavin; 2, cell free extract of pRFN4 in *E. coli* 43R; 3, cell free extract of pRFN4-Pp *lux*LP-N-*lum*P in *E. coli* 43R. Lum, lumazine; Rf, riboflavin.

**Figure 10 ijms-24-05096-f010:**
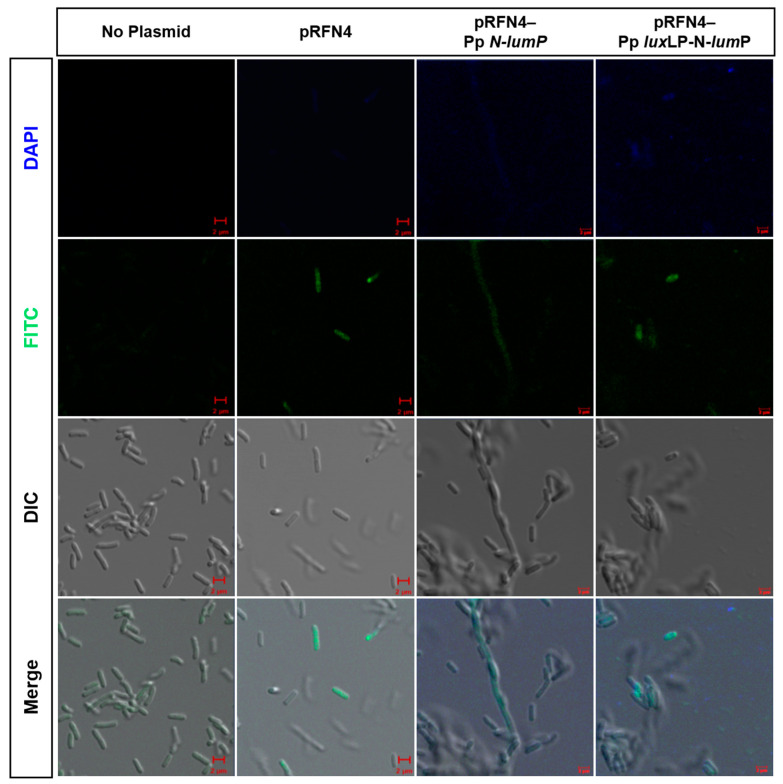
Fluorescence microscopic images of *E. coli* 43R containing the recombinant plasmids of pRFN4. *E. coli* 43R itself (No plasmid), transformed with pRFN4, with pRFN4-Pp N-*lum*P, and with pRFN4 Pp *lux*LP N-*lum*P. DAPI, Diamidino-2-phenylindol; FITC, Fluorescein isothiocynate; DIC, Differential interference contrast. Scale bar; 2 µm.

**Table 1 ijms-24-05096-t001:** Bacterial strains and plasmids used in this study.

**Strains**	**Characteristics**	**Source**
*E. coli* XL-1 Blue	Cloning strain	Real Biotech Corporation
*E. coli* 43R	Mutant of *E. coli* RR1 strain	Miyamoto.et al. (1987) *J. Bacteriol.* [16]
**Plasmids**	**Characteristics**	**Source**
pRFN4	Recombinant plasmid containing the riboflavin genes from *Bacillus subtilis*	Illarionov et al. (2004) *J. Org. Chem.* [13]
pT7-5 PpEE	Recombinant pT7-5 plasmid containing the *lux*L *and lux*C genes, as well as their intergenic DNA from *P. phosphoreum* NCBM 844	Woo et al. (2005) *Kor. J. Microbiol.* [14]
pRFN4- Pp N-*lum*P	pRFN4 plasmid inserting the gene of 5′-terminal half of *lux*L for N-LumP from *P. phosphoreum*	This study
pRFN4- Pp *lux*CP	pRFN4 plasmid inserting the promoter region for *lux*C from *P. phosphoreum*	This study
pRFN4- Pp *lux*CP-N-*lum*P	pRFN4-Pp N-*lum*P plasmid inserting the promoter region for *lux*C from *P. phosphoreum*	This study
pRFN4- Pp *lux*LP-N-*lum*P	pRFN4-Pp N-*lum*P plasmid inserting the DNA for promoter region for *lux*L from *P. phosphoreum*	This study

**Table 2 ijms-24-05096-t002:** Oligonucleotide primers for PCR amplification of the *lux* promoter region. Restriction sites for *EcoR*I (GAATTC) and *Bam*HI (GGATCC) are underlined.

Species	Primer	Sequence
*P. phosphoreum*	N-*lum*P forward	5′-CAAATGAAGAAGAATTCCATTTTAAATATG-3′
*P. phosphoreum*	N-*lum*P reverse	5′-AAACTTTAAGAGGATCCTCTTCTTC-3′
*P. phosphoreum*	*lux*C promoter forward	5′-CTTTATCATAGAATTCCCTTTAAATTCC-3′
*P. phosphoreum*	*lux*C promoter reverse	5′-CTTGTTATTTTTAGAATTCTCTTATTCTTTC-3′
*P. phosphoreum*	*lux*L promoter forward	5′-GTTTATCGAATTCCATTTTAAATCATC-3′
*P. phosphoreum*	*lux*L promoter reverse	5′-CATATTTAAAATGGAATTCTTCATTTGG-3′

## Data Availability

Not applicable.

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
