# Peer review of "Development of Fluorescent Bacteria with *Lux* and Riboflavin Genes"

_ijms, 2023, doi:10.3390/ijms24065096_

Round 1
Reviewer 1 Report
The communication appears to be interesting and technically sound. The authors have represented quality experimentations, though there are some minor modifications needed for proper readability of the same. The comments are listed below:
1. The result portion appears to be monotonous. The authors may sub-sectionize the result portion with proper headings for better readability.
2. The authors have stated the possibility of usage of fluorescent bacteria to construct biosensor. Please add a brief section in discussion regarding how this kind of study can be translated towards developing a biosensor.
3. As mentioned by the authors, Ribulose is utmost necessary towards formation of riboflavin. Have the authors checked the fluorescenec intensity of the transformed bacteria under carbohydrate stress condition?
4. What would be the consequences if the authors avoid intergenic DNA inserted between promoters of LuxL and LuxC? Have they performed the experiment without the intregenic DNA? please discuss briefly.
5. Provide high quality figures for better readability.
Author Response
Thanks a lot for your careful reading and critical comments. I attached the file containg the reponces to your suggestions and directions.

Reviewer 2 Report
The paper by Lim et al., entitled “Development of Fluorescent Bacteria with lux and Riboflavin Genes” is a study on fluorescent bacterial system with lux and riboflavin genes. The authors developed fluorescent bacteria. The methods and the data presented here are generally sound. This study might be of interest of readers of this journal. However, this reviewer hopes that the authors consider the following points.
1. In Figure 7 and 8: The authors did not directly confirm the production of lumazine, riboflavin, and lumazine protein in each bacteria. To make authors’ discussion more reliable, the authors should show the production amount of lumazine, riboflavin, and lumazine protein in each bacteria by LCMS analysis, SDS-PAGE analysis, or similar analysis.
2. In Figures 3-5 and 7-9: The resolution of Figures 3-5 and 7-9 should be improved.
Author Response
Thanks a lot for your critical readings and valuable suggestions. I attached file containg the responces for your to your request and directions.

Reviewer 3 Report
The article is devoted to the construction of brightly fluorescent E.coli bacteria using lumazine and N-terminus of lumazine protein. The work is noteworthy, but there are several comments.
1. It is very difficult to read the names of plasmids containing the preposition "of". Such name occuring in a sentence may confuse a reader. For example, in the phrase “…plasmids were constructed by amplifying the DNA encoding the N-LumP gene (luxL) of Photobacterium phosphoreum (Pp N-lumP of
pRFN4)…”, the reader may think that there is an explanation in brackets, meaning that the fragment “Pp N-lumP” was amplified from the pRFN4 template.
2. Usually in the literature hybrid plasmids names begin with a small letter “p”, then the name of the plasmid is written in a continuous line. For example: pT7-5 or pRFN4 are ordinary names. The names of plasmids (at least those made in this work) should be changed whenever it is possible.
3. The previously quoted phrase “N-LumP gene” is not true, because N-LumP is a protein name. The sentence should be fixed. For example: “…amplifying the DNA containing the 5'-terminus of luxL gene (encoding N-LumP) of Photobacterium phosphoreum” or “….amplifying the DNA encoding the N-LumP (5'-terminus of luxL gene) of Photobacterium phosphoreum".
4. Line 80 “…mutated riboflavin synthase gene (ribB)…” confuses me, I don’t understand it. Riboflavin synthase gene is ribE gene. What gene is actually mutated in pRFN4??? What is the mutation: deletion, frameshift, or nucleotide substitution? This should be briefly described.
5. On the figure 3, the symbols confuse me. I believe that bars represent the absence of aminoacid in the sequence, but in the end of RibE protein bars indicate some aminoacid or several aminoacids instead? Check carefully all the figures with protein sequences and symbols on them.
6. On the fig 5 second caption from bottom “N-lump” in the plasmid name should be changed to “N-lumP”.
7. Generic and specific names should be written in italics, even in the bibliography.
8. Materials and Methods come after Discussion, so the numbers of the references should be changed in the order of their occurence.
9. The last phrase in "Discussion" can be strengthened, because it seems like this type of biosensor can be extended by using inducible promoters to the gene encoding N-LumP. Such biosensors can be developed based on both E. coli and B. subtilis by analogy with [doi: 10.1016/S0065-2911(04)49003-1; doi: 10.3390/ijms22179571].
Author Response
Thanks a lot for your critical reading valuable suggestions. I attached the file containg the reponces to your critical comments and directions.

Round 2
Reviewer 1 Report
In the revised manuscript, the authors have addressed most of the suggestions in a very satisfactory way. The manuscript can now be accepted for publication.
Reviewer 2 Report
The revision from the authors addresses my concerns, and this reviewer appreciates the authors’ efforts. However, this reviewer hopes that the authors consider the following points.
1. In Figure 9c: The authors should describe the procedure of TLC analysis including the development solvent in the section of “Materials and Methods."
2. In Figure 7: Does the amount of lumazine protein affect the fluorescence intensity? If so, the authors should show the amount of lumazine protein in each bacterium.
Author Response
Thanks alot for your valuable suggestions.
- We described the procedure of TLC in Figure 7C in the section of "materials and Methods"
- In Figure 7, we did not checked the expression of the gene for N-LumP. We explained in detail in attached file. I sincerely hope that you understand this situations.
